# *Buxus* and *Tetracentron* genomes help resolve eudicot genome history

Andre S. Chanderbali [1✉], Lingling Jin[2], Qiaoji Xu[3], Yue Zhang[3], Jingbo Zhang[4], Shuguang Jian[5], Emily Carroll[6], David Sankoff [3], Victor A. Albert [6], Dianella G. Howarth[4], Douglas E. Soltis [1,7,8,9] & Pamela S. Soltis [1,8,9]

Ancient whole-genome duplications (WGDs) characterize many large angiosperm lineages, including angiosperms themselves. Prominently, the core eudicot lineage accommodates 70% of all angiosperms and shares ancestral hexaploidy, termed *gamma*. Gamma arose via two WGDs that occurred early in eudicot history; however, the relative timing of these is unclear, largely due to the lack of high-quality genomes among early-diverging eudicots. Here, we provide complete genomes for *Buxus sinica* (Buxales) and *Tetracentron sinense* (Trochodendrales), representing the lineages most closely related to core eudicots. We show that *Buxus* and *Tetracentron* are both characterized by independent WGDs, resolve relationships among early-diverging eudicots and their respective genomes, and use the RAC-CROCHE pipeline to reconstruct ancestral genome structure at three key phylogenetic nodes of eudicot diversification. Our reconstructions indicate genome structure remained relatively stable during early eudicot diversification, and reject hypotheses of *gamma* arising via interlineage hybridization between ancestral eudicot lineages, involving, instead, only stem lineage core eudicot ancestors.

[1] Florida Museum of Natural History, University of Florida, Gainesville, FL, USA. [2] Department of Computer Science, University of Saskatchewan, Saskatoon, SK, Canada. [3] Department of Mathematics and Statistics, University of Ottawa, Ottawa, ON, Canada. [4] Department of Biological Sciences, St. John's University, Queens, NY, USA. [5] South China Botanical Garden, Chinese Academy of Sciences, Guangzhou, China. [6] Department of Biological Sciences, University at Buffalo, Buffalo, NY, USA. [7] Department of Biology, University of Florida, Gainesville, FL, USA. [8] Biodiversity Institute, University of Florida, Gainesville, FL, USA. [9] Genetics Institute, University of Florida, Gainesville, FL, USA. ✉email: achander@ufl.edu

Flowering plants (angiosperms), with nearly 400,000 species and a fossil record that dates to the Early Cretaceous, have a complex evolutionary history marked by early and rapid lineage divergences[1–3]. Whole-genome duplication (WGD) events have also been frequent in angiosperms, and indeed all extant species are ancient polyploids descended from a common ancestor that experienced at least one WGD[4,5]. Subsequent polyploidy events have been identified throughout angiosperm phylogeny, often coinciding closely with the origin and/or radiation of major clades[6–10]. Notably, the core eudicots (Gunneridae[11]), nested in the eudicot clade, descend from an ancient hexaploid formation, termed *gamma*[12–15], and account for ~70% of extant angiosperm species. Moreover, a novel suite of floral features, 'whorled pentamery' with flower parts arranged in concentric whorls of five[16–18], evolved shortly after the origin of the core eudicots[11,19] and could be genetically linked to this ancient hexaploidy event, e.g., through multiplications or rearrangements of floral transcriptional regulators[15]. Such a causal relationship between *gamma* and whorled pentamery, although still speculative, is consistent with the widely acknowledged role of gene and genome duplications providing the genetic raw material for evolutionary innovation[9,20].

The phylogenetic timing and mechanism of *gamma* hexaploidy are currently unresolved. Hypotheses on the topic mostly envision a two-step process, in which the product of an initial WGD fused with a third genome in a second polyploidization, possibly via a wide cross after an extended period of random fractionation (loss of either copy of duplicated genomic regions following WGD) in the tetraploid intermediate[21]. The breadth of this putative wide cross is also unclear and possibly includes extant early-diverging eudicot lineages[13,15,22]. Alternatively, one of the *gamma* subgenomes may have been more resistant to fractionation, and all three subgenomes may have been joined rapidly in evolutionary time[21], perhaps in an autohexaploidy event[23]. It has also been argued that *gamma* hexaploidy derives from an initial tetraploidy shared by all eudicots[24,25]. Further still, the lack of clear evidence of *gamma* outside of the core eudicots may be due to stochastic gene loss over more than 100 million years of independent evolution[23]. Efforts to evaluate evolutionary scenarios of *gamma* origins have been hampered by limited data and unsettled sister-group relationships to the core eudicots. Plastome sequence data support either Buxales[19,26,27] or Trochodendrales[28,29] as immediate sisters to the core eudicots, while single-copy nuclear (SCN) genes from transcriptome data sets have recovered a Buxales+Trochodendrales clade placed sister to the core eudicots[15,30]. Thus, despite considerable research interest, the timing and mechanism of *gamma* formation have remained unresolved.

We here provide genome assemblies for *Buxus sinica* (Buxales) and *Tetracentron sinense* (Trochodendrales), which represent, either individually or collectively, the sister lineage of core eudicots[15]. These two genome assemblies complement those available for other early-diverging eudicot lineages[22,31–33] and permit evaluations of eudicot phylogeny and *gamma* origins based on phylogenomics, molecular evolution, and synteny. In addition, we employ the RACCROCHE[34] pipeline of algorithms to infer the ancestral genomes at three sequential nodes of the eudicot radiation.

## Results and discussion

**Genome assembly, annotation, and structure.** Chromosome-scale nuclear genome assemblies for *Buxus* and *Tetracentron* were produced from PacBio long-read contigs assembled with the FALCON/FALCON-unzip pipeline[35] and scaffolded by Hi-C technology[36] (Fig. 1; Supplementary Data 1). The *Buxus* assembly totals 764 Mb (90% of the estimated genome size of 850 Mb), with 7180 contigs (N50 = 164 kb) in 63 scaffolds (N50 = 56 Mb), of which 14 contain 763 Mb (99.8%) of the assembly. The *Tetracentron* assembly totals 908 Mb (93% of the estimated genome size of 975 Mb), with 6178 contigs (N50 = 238 kb) in 662 scaffolds (N50 = 54 Mb), of which 19 contain 856 Mb (94.5%) of the assembly. The largest 14 and 19 scaffolds of the *Buxus* and *Tetracentron* assemblies, respectively, correspond with the known chromosome numbers of these taxa[37,38]. Benchmarking Universal Single-Copy Orthologs (BUSCO) analyses[39,40] estimate 96.3% and 93.5% completeness for the *Buxus* and *Tetracentron* genomes, respectively (Supplementary Data 2). Transposable elements and other repeat sequences account for 76.4% and 78.5% of the *Buxus* and *Tetracentron* assemblies, respectively (Supplementary Data 3). In *Buxus*, LTR retrotransposons (26.8%), followed by LINEs (4.9%) and DNA transposable elements (2.8%), are most abundant, with Ty3/*Gypsy* and Ty1/*Copia* retrotransposons accounting for 87.2% and 13.0% of the LTRs, respectively. LTRs (27.4%), LINES (4.6%), and DNA transposable elements (2.9%) account for most of the *Tetracentron* repeats, with Ty3-*Gypsy* (62.6%) and Ty1/*Copia* (36.6%) retrotransposons best represented among the LTRs. Annotation of the repeat-masked assemblies yielded 27,027 and 30,704 protein-coding gene models, including 86.9% and 80.5% of the BUSCO genes, in *Buxus* and *Tetracentron*, respectively (Supplementary Data 2). Our *Tetracentron* assembly is similar to one produced for another individual of this species[33] in terms of BUSCO statistics and annotation metrics, but differs in size (908 vs 1170 Mb) and the number of chromosome-size scaffolds (19 vs 24). We are unable to account for these differences, but our assembly closely matches the genome size measured by flow cytometry, and the only reported chromosome count of $n = 24$[41] for *Tetracentron* has been discredited[37].

Analyses of synonymous changes per synonymous site (*Ks*) and intragenomic synteny indicate that *Buxus* and *Tetracentron* are both paleopolyploids, with one and two rounds of WGDs in their respective evolutionary histories. *Buxus* syntenic paralogs (paleologs) constitute extensive blocks of colinear genome sequence across pairs of chromosomes and are characterized by *Ks* values close to 1.0 (Fig. 1c). *Ks* values for *Tetracentron* paleologs are concentrated near *Ks* = 0.5, but colinear genome sequences are distributed among four chromosomes (Fig. 1d), together suggesting two WGDs in close succession. The two *Buxus* subgenomes are highly conserved, with synteny blocks that often extend across much of the whole chromosomes, while the four subgenomes of *Tetracentron* appear to be highly rearranged at the chromosomal level (Fig. 1). The extent to which this structure reflects genome reshuffling, which is a prominent mechanism of post-polyploid diploidization (PPD) after WGDs[42], or artifacts of genome assembly, is unclear. In favor of PPD processes, the *Tetracentron* genome is appreciably downsized compared to its sister species, and the only other living member of Trochodendraceae, *Trochodendron aralioides* (0.9 versus 1.6 GB), which shares two WGDs with *Tetracentron*[33] but exhibits more extensive blocks of inter-chromosomal synteny (Supplementary Fig. 1).

**Phylogenetic positions of *Buxus* and *Tetracentron*.** To reconstruct the branching sequence of the early eudicot radiation, we analyzed phylogenetic data sets for representative angiosperms composed of hundreds of BUSCO genes[43], the Angiosperms353 loci[44], and orthogroups identified de novo by the Orthofinder pipeline[45]. Coalescence-based analyses of all three data sets place Ranunculales as sister to all other living eudicot lineages, with Proteales (including Sabiaceae) diverging next, and a Buxales +Trochodendrales clade as sister to the core eudicot clade (Fig. 2a;

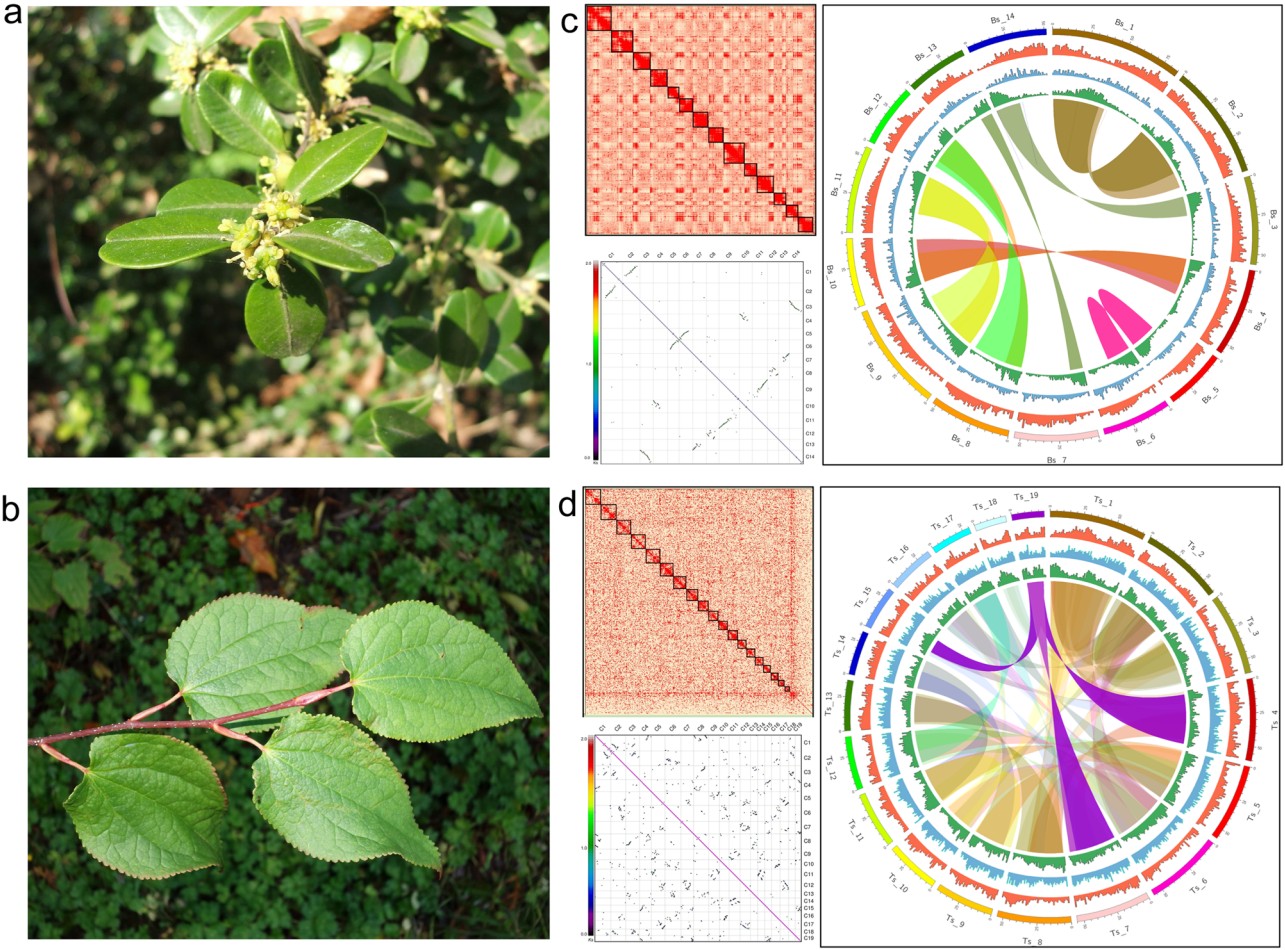

**Fig. 1 Habit and genome assembly features of *Buxus* and *Tetracentron*.** Flowering branch of *Buxus sinica* (**a**) courtesy of PiPi; and leafy shoot of *Tetracentron sinense* (**b**) courtesy of Daderot. Hi-C contact heatmaps, intragenomic synteny with syntenic blocks colored according to the *Ks* scale, and Circos plots for *Buxus* (**c**) and *Tetracentron* (**d**). Concentric tracks in the Circos plots, from innermost outwards, show gene, *Copia*, and *Gypsy* retrotransposon densities per 1 Mb, and chromosomes, while ribbons connect inter-chromosomal syntenic regions. Source data underlying Fig. 1c, d are provided as a Source data file.

left panel, Supplementary Figs. 2a and 3). Concatenated data sets of the SCN loci, whether analyzed in Maximum Likelihood (Fig. 2a, right panel, Supplementary Fig. 2b) or Bayesian Inference (Supplementary Fig. 4) frameworks, recover Buxales alone as the core eudicot sister group, with Trochodendrales as sister to this Buxales +core eudicot clade. Although this branching sequence receives maximal statistical support in both Maximum Likelihood (boot-strap) and Bayesian Inference (posterior probability) analyses, incomplete lineage sorting (ILS) is a potential confounding factor in phylogenetic analyses of concatenated data sets in the face of rapid radiations[46], as is the case for the eudicots. Indeed, the quartet-support values associated with the Buxales+Trochoden-drales clade in the coalescence tree indicate considerable gene tree discordance with respect to the positions of these taxa. Further exploration of conflicts affecting the eudicot clade, visualized as a cloudogram of gene trees (Fig. 2b), however, reveals that ~30% of the gene trees support the Buxales+Trochodendrales clade, while only ~18% support either Buxales or Trochodendrales as the core eudicot sister group (Supplementary Data 4). We also estimated the branching sequence of early-diverging eudicots using the 'Trees in the Peaks' method, which reconstructs speciation and poly-ploidization events from *Ks* and similarity score distributions of syntenic homologs[47,48] (Fig. 2c). This method, which requires that ancestral *Ks* and similarity scores and/or their ranges must precede (greater *Ks* or lower similarity) or overlap those in the descendants,

was applied to evaluate each of all possible binary rooted phylo-genies. The only branching sequence that satisfies these conditions is one in which Buxales and Trochodendrales are collectively sister to the core eudicots. Specifically, the peak *Ks* value of syntenic orthologs that diverged via the *Buxus/Tetracentron* speciation is younger than those derived from the phylogenetic divergence of *Vitis* (a core eudicot) from *Buxus* or from *Tetracentron*.

**Phylogenomics of eudicot subgenomes**. Synteny-guided phylo-genomic analyses of eudicot subgenomes were conducted to assess the several hypothesized scenarios for the origin of *gamma* hexaploidy (Fig. 3). Pairwise analyses of inter-genomic colli-nearity (macrosynteny) and fractionation patterns identify extensive regions of early-diverging eudicot genomes shared with the *gamma*-derived hexaploid genome of *Vitis*, and each other (Supplementary Figs. 5–9). The ratios of syntenic depths (the number of times a genomic region is syntenic to regions in another genome) in these comparisons reflect the number of subgenomes, or level of ploidy, for the respective species. Thus, we see 2:3 syntenic depth between *Buxus* and *Vitis*, and 4:3 syn-tenic depth between *Tetracentron* and *Vitis*, while *Tetracentron* to *Buxus* is 4:2 in syntenic depth. Likewise, as previously reported, *Aquilegia* and *Nelumbo* each exhibit 2:3 syntenic depth with *Vitis*, and 2:2 with each other. Collectively, these macrosyntenic alignments approximate the modern distribution of the seven

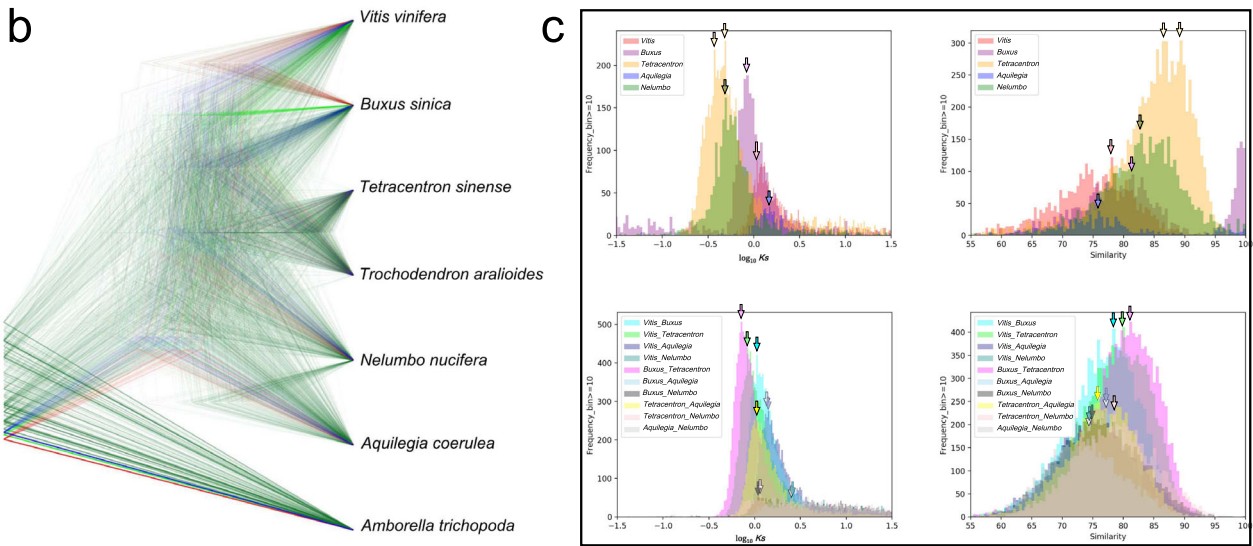

**Fig. 2 Phylogenetic relations of *Buxus* and *Tetracentron*. a** Phylograms depicting the coalescent solution of individual Maximum Likelihood (ML) gene trees (left) and partitioned ML analysis of a supermatrix of nucleotide sequence alignments (right). Node labels indicate quartet (coalescence) and bootstrap (supermatrix) support values, and orange stars highlight the positions of Buxales and Trochodendrales in the two trees. **b** Cloudogram of 763 SCN gene trees illustrating discordance surrounding the deep branches of eudicot phylogeny. The most frequent trees are blue, the next most frequent red, the third most frequent green, and the rest are dark green. **c** *Ks* (left) and Similarity (right) distributions showing peaks (arrows) that stem from WGD (top) and speciation events (bottom), respectively. Source data underlying Fig. 2b, c are provided as a Source data file.

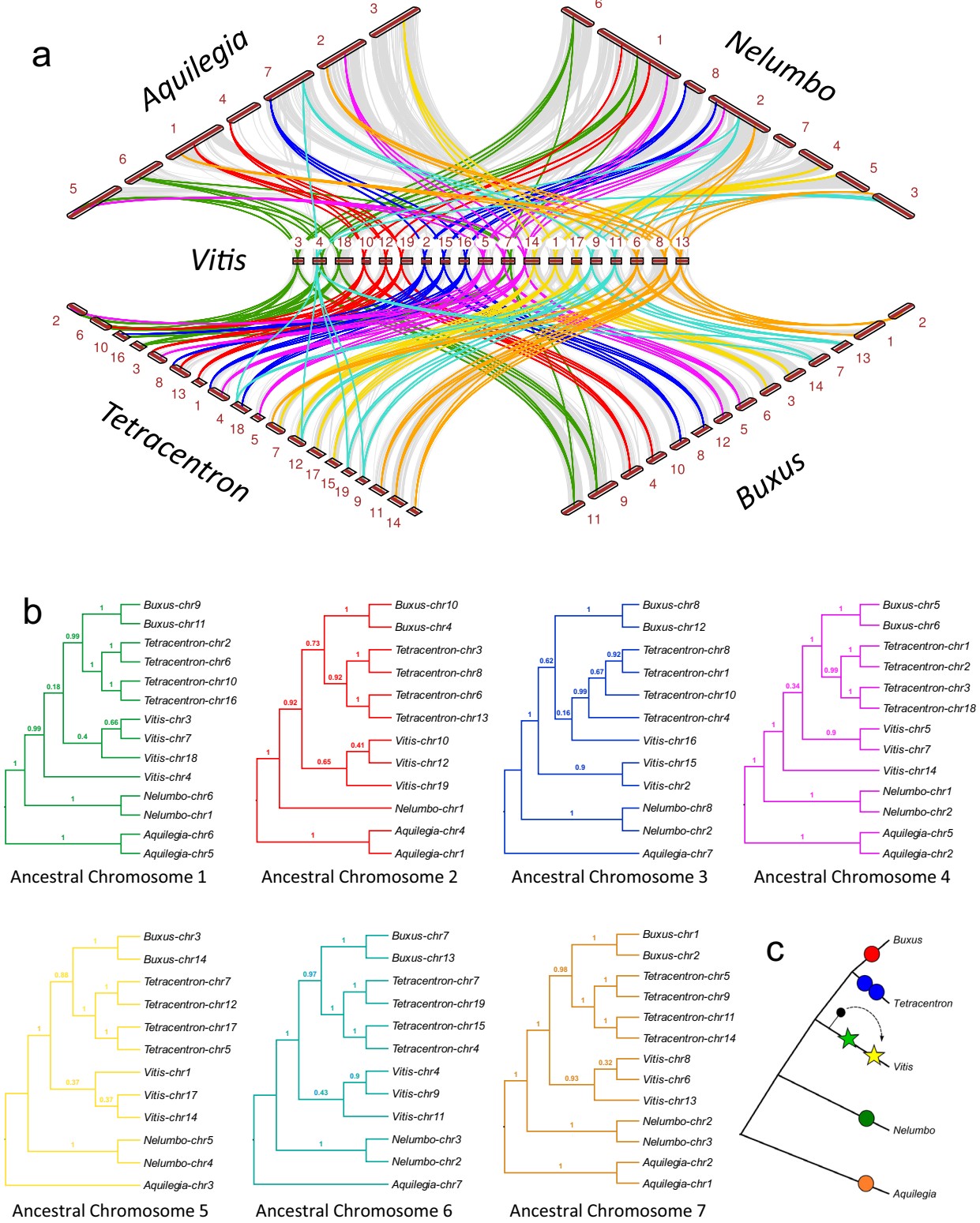

**Fig. 3 Synteny and phylogenomics of eudicot subgenomes. a** Macrosyntenic alignments of early-diverged eudicots against *Vitis* with tracking of genomic positions by color-coded syntenic blocks representing the seven ancestral eudicot chromosomes. **b** Coalescence-based phylogenies of syntelogs derived from duplication events affecting the seven ancestral eudicot chromosomes. Green, red, blue, purple, yellow, aqua, and brown tracks highlight positions of ancestral chromosomes 1 through 7, respectively. Branch labels are posterior probabilities. **c** Schematic reconstruction of ancient eudicot WGD history. Differently color-filled circles label putative independent duplication events and stars highlight the two *gamma* WGDs in which the third genome is donated to the initial tetraploid (green star) from an extinct lineage to form the hexaploid (yellow star). Source data underlying Fig. 3a are provided as a Source data file.

ancestral eudicot chromosomes (Fig. 3a, Supplementary Data 5, and see below), the evolutionary histories of which we have estimated through phylogenetic analyses of 1932 gene trees populated with 15872 genes (Fig. 3b). For example, syntenic blocks descended from ancestral chromosome 4 (purple tracks in Fig. 3a) occupy regions of *Vitis* chromosomes 5, 7, and 14, as well as portions of chromosomes 2 and 5 of *Aquilegia*, 1 and 2 of *Nelumbo*, 5 and 6 of *Buxus*, and 1, 2, 3, and 18 of *Tetracentron*. Microsynteny (gene level) alignments within these major synteny blocks comprise 235 homologous loci and a total of 1837 syntelogs (genes derived from the same ancestral genomic region) useful for inferring the evolutionary history of ancestral chromosome 4 (see Supplementary Data 5 for the modern distribution and representation of each ancestral chromosome). The coalescent solution of phylogenetic trees for these 235 loci shows that duplicated blocks of ancestral chromosome 4 now present in *Aquilegia*, *Nelumbo*, *Buxus*, and *Tetracentron* constitute lineage-specific clades (Fig. 3b), indicating that ancestral chromosome 4 was duplicated independently in each of the respective stem lineages of these four modern genomes. Indeed, the duplicated blocks of all seven ancestral chromosomes in *Aquilegia*, *Nelumbo*, *Buxus*, and *Tetracentron* constitute lineage-specific groupings (Fig. 3b), providing consensus that their respective WGDs are independent events and, importantly, exclusively involved genome donors that belonged to their respective clades, i.e., their stem lineage ancestors.

Phylogenetic alliances of the seven ancestral chromosomes occupying the modern, *gamma*-derived, *Vitis* genome are less clear. Of the three copies of ancestral chromosome 4, the syntenic blocks preserved on *Vitis* chromosome 5 and 7 form a well-supported sister group, but the block on *Vitis* chromosome 14 is placed as an earlier branch, albeit with low support. *Vitis*-specific clades were also not recovered for ancestral chromosomes 1 and 3, although again without high statistical support for non-monophyly. However, triplicated copies of ancestral chromosomes 2, 5, 6, and 7 in the *Vitis* genome group together as each other's closest relatives. Although clade support is strong only for the copies of ancestral chromosome 7 currently preserved on *Vitis* chromosomes 6, 8, and 13, the phylogenies of these four sets of genomic regions suggest they uniquely share a common ancestor, one that evolved separately from the other, earlier-diverged, eudicot lineages. Altogether, we recover *Vitis*-specific groupings for duplicates of four of the ancestral eudicot chromosomes, albeit as a well-supported clade only once. The relationships of the other three ancestral chromosomes may best be described as phylogenetically unresolved. Importantly, these findings are inconsistent with evolutionary scenarios of *gamma* formation through an extremely wide cross between a core eudicot and an early-diverging eudicot lineage, as has been previously proposed[22]. An initial tetraploidy event in the common ancestor of the eudicots[24] is also inconsistent with our finding that paralogous genomic blocks in *Aquilegia*, and all other basal eudicots, constitute lineage-specific clades. The only evolutionary scenario consistent with our analyses is one in which *gamma* hexaploidy exclusively involved stem lineage ancestors of extant core eudicot species as genome donors. As such, if hexaploidy was attained via a two-step process of sequential WGDs, the third of the *gamma* genomes must have been donated from a now extinct lineage that branched off the core eudicot ancestral line before the initial tetraploidy event (Fig. 3c).

**Ancestral genomes**. The independence of each of the WGD events associated with each of the early-diverging eudicot lineages implies unduplicated ancestral genomes leading all the way from the ancestral angiosperm up to *gamma* and the core eudicots.

We explore this key inference through ancestral genome reconstruction. We reconstructed ancestral genomes at three nodes of the eudicot phylogeny (Fig. 4a): the common ancestor of the core eudicot clade (ancestor 3), two sequentially older nodes ancestral also to *Buxus* and *Tetracentron* (ancestor 2), and *Nelumbo* (ancestor 1).

All three of these ancestral genomes are reconstructed as seven putative protochromosomes, each with between 700 and 1600 protogenes, totaling more than 8000 protogenes, arranged in their ancestral order (Fig. 4b). Our ancestral genome reconstructions include ~2000 more (ca. 25%) ordered protogenes than previous reconstructions of an ancestral eudicot genome[49]. To understand the early evolution of eudicot genome structure, we partitioned the modern eudicot chromosomes into sets of syntenic regions and painted each of these according to its corresponding protochromosomes (Fig. 4c; Supplementary Fig. 10). These projections relate modern eudicot genomes to successive ancestral precursors and provide insights into the relative timing of any structural changes during eudicot genome evolution. Projections of the three ancestral genome reconstructions onto *Vitis* chromosomes (Fig. 4c) are globally similar, indicating genome structure remained relatively stable during early eudicot diversification. Inconsistent with the hypothesis of one ancestral eudicot tetraploidy[24], these projections indicate that fusion of the two ancestral chromosomes now combined in *Vitis* chromosome 7 and *Aquilegia* chromosome 5 (juxtaposed purple and green blocks in Fig. 4c and Supplementary Fig. 10, respectively) did not occur prior to the origin of the eudicot ancestor. Were this the case, both sections of these *Vitis* and *Aquilegia* chromosomes would be painted with a common color representing one ancestral chromosome whose 'chimeric' origin would be invisible to our methods. Instead, these, and other, chromosomal fusions appear to be independent, lineage-specific events that post-date ancestral genome arrangements. Several other genomic rearrangements, as measured by the 'choppiness' of chromosomal paintings (Supplementary Data 6), emerge from our reconstructions. In the case of *Vitis*, the modern genome has accumulated 41 inter-chromosomal exchanges relative to ancestors 1 and 2, and 31 after ancestor 3. The reduced number of inter-chromosomal exchanges indicates greater similarity of *Vitis* to the core eudicot ancestor (ancestor 3) relative to the more ancient ancestors 1 and 2. A similar reduction of inter-chromosomal exchanges, from 67 (relative to ancestor 2) to 56 (relative to ancestor 3), was also observed for *Amaranthus tuberculatus*, the other core eudicot genome in our analyses. As such, we can reject the occurrence of any single WGD in the eudicot stem lineage and instead firmly resolve independent WGDs in each modern eudicot lineage, including the core eudicots with their unique *gamma* hexaploid structure.

Our *Buxus* and *Tetracentron* genome assemblies have facilitated rigorous assessments of alternative hypothesized scenarios for the origin of *gamma*, a key hexaploidy associated with a major event in the history of terrestrial life, the origin of core eudicots, which comprise the vast majority of flowering plants. We have presented and analyzed several lines of evidence, including *Ks* distributions, genomic synteny, fractionation bias, phylogenomics, and ancestral genome reconstruction, that bear relevance to the phylogenetic and WGD history of the early-diverging eudicot angiosperms. These analyses reconstruct the sequential branching order of the initial eudicot radiation and show that each of the early-diverging eudicot lineages is characterized by its own independent duplication event(s). We find no evidence to support hypotheses that a single polyploidy event might have been formative for eudicot diversification as a whole. Instead, our analyses place *gamma* hexaploidy on the stem lineage of core eudicots and rule out a role for other living early-diverging

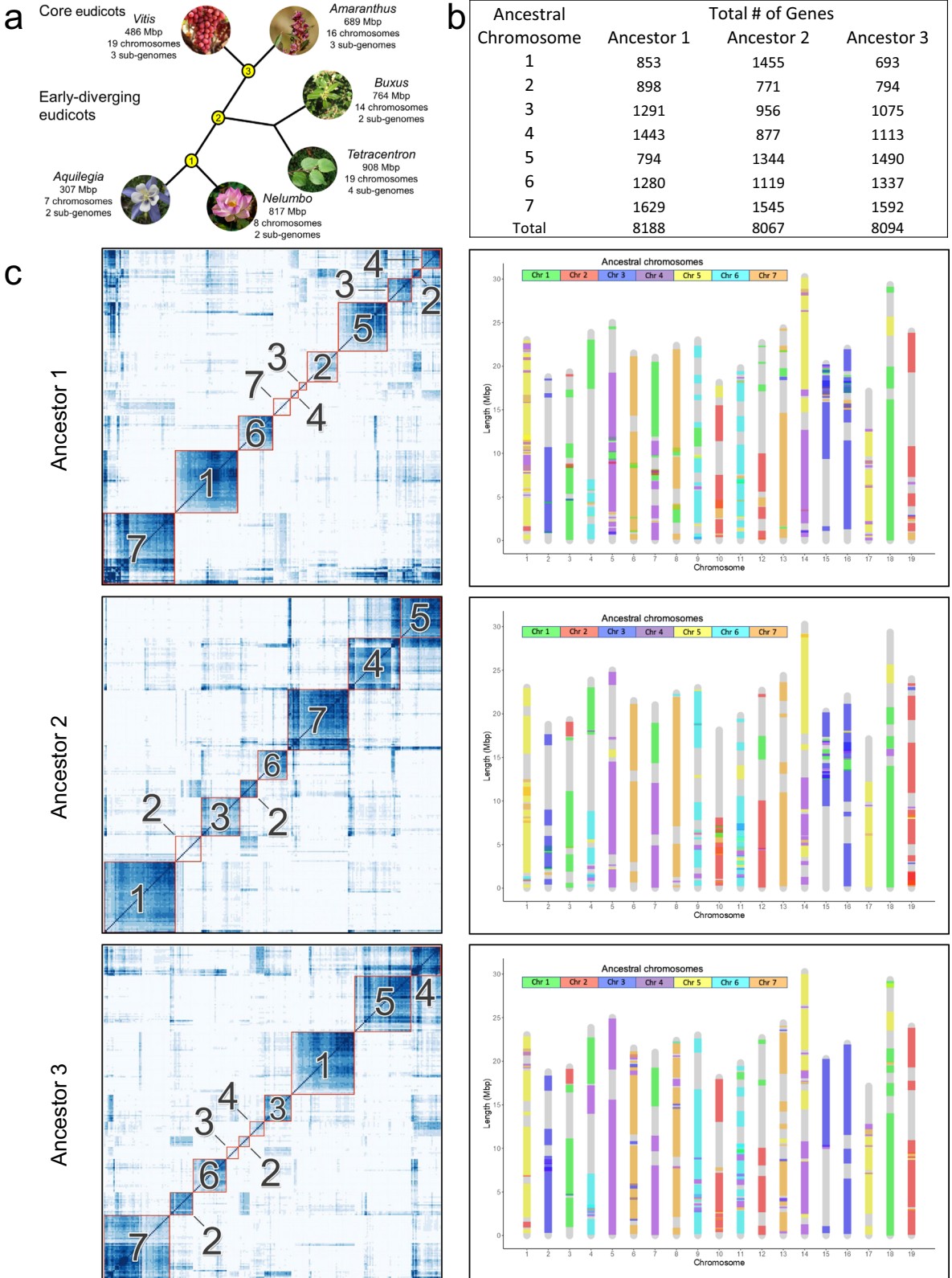

**Fig. 4 Ancestral eudicot genomes. a** Schematic phylogeny of the eudicot clade depicting extant and ancestral genomes (nodes 1–3) examined here. Images Credits: *Vitis*, Bob Nichols; *Amaranthus*, Patrick Alexander; *Buxus*, PiPi; *Tetracentron*, Daderot; *Nelumbo*, Engin Akyurt; *Aquilegia*, Ejohnsonboulder; all available in the Public Domain via Wikimedia Commons. **b** Protogene content of ancestral genomes. **c** Heatmaps of conserved synteny supporting the delimitation of seven protochromosomes in each ancestral genome (left panel), and *Vitis* chromosomes painted according to the protochromosomes of the three diploid ancestral genomes (right panel). Source data underlying Fig. 4c are provided as a Source data file.

eudicots as genome donors, a possibility that was consistent with the results of previous analyses[13–15,22]. Without a single, linking WGD common to all eudicots, an argument that one polyploidy event may have helped spur the massive eudicot diversification (via adaptive, alternative deployments of duplicate genes), even following a time lag, is not supported by our data. Instead, each independent WGD among the early-diverging eudicot lineages, other than *gamma*, underlies relatively species-poor lineages that show limited fossil or living evidence for extensive radiation. Thus, with the genomes of all living early-diverging eudicot lineages now examined for a possible genomic contribution to *gamma*, the origin of *gamma* remains another abominable angiosperm mystery despite intensive study.

## Methods

**DNA extraction, sequencing, and assembly**. *Buxus sinica* and *Tetracentron sinense* tissues were obtained from individuals cultivated at the University of Wisconsin-Madison (accession no. UW 136) and the University of Washington Arboretum, Seattle (accession no. 385-62), respectively. Genome sizes for these accessions were estimated using flow cytometry with BD CellQuest Pro software (Supplementary Data 7) by the Benaroya Research Institute (Seattle, WA). High-molecular-weight genomic DNA was isolated from young leaf tissue using modified nuclei-preparation and cetyltrimethylammonium bromide (CTAB) DNA extraction methods. Briefly, leaf tissue was ground to a fine powder under liquid nitrogen and mixed with nuclear isolation buffer (15 mM Tris, 10 mM EDTA, 130 mM KCl, 20 mM NaCl, 1 mM Spermine, 1 mM Spermidine, 8% PVP-10, 0.1% Triton X-100, and 7.5% 2-mercaptoethanol), passed sequentially through 100 and 40 μm mesh filters, treated with 1% Triton X-100, and centrifuged at $2000 \times g$ for 10 min at 4 °C to pellet the nuclei. The pellet resuspended for 1 h at 65 °C in lysis buffer (100 mM Tris-HCl, 100 mM NaCl, 50 mM EDTA, 2% CTAB, 1% PEG 6000), and high-molecular-weight DNA was isolated from the lysate via 24:1 chloroform/isoamyl alcohol and purified with the QIAGEN Genomic kit. SMRTbell 20-kb libraries were generated and sequenced on the PacBio RSII platform to ~160x genomic coverage. In addition, Hi-C libraries were prepared and sequenced to coverage depths of ~40x by Phase Genomics (Seattle, WA). PacBio reads were assembled using the pb-assembly suite of programs which includes the FALCON/FALCON-unzip assembly pipeline and performs contig phasing and polishing[35]. The polished assemblies were deduplicated with Purge Haplotigs[50] and scaffolded using Proximity Guided Assembly (PGA) and Hi-C reads by Phase Genomics (Seattle, WA).

**RNA-seq data**. Transcriptome assemblies were produced for *Buxus sinica* and *Tetracentron sinense* to aid annotation of their genome assemblies. We also produced transcriptome assemblies for six additional early-diverging eudicots (*Buxus sempervirens, Meliosma dillenifolia, Nelumbo lutea, Sabia emarginata, Sabia swinhonei, Trochodendron aralioides*), as well as the core eudicot (*Gunnera manicata*), to improve taxon sampling in phylogenetic analyses. Paired-end RNA-seq libraries were constructed from polyA selected total RNA extracted from floral and/or leaf tissues (Supplementary Data 8), and sequenced using the Illumina HiSeq 3000 system. Reads were trimmed with Trimmomatic[51] and assembled using Trinity[52]. Coding DNA (CDS) and protein sequences were predicted with TransDecoder (http://transdecoder.github.io).

**Annotation**. Genomes were annotated using the MAKER pipeline[53]. De novo transcriptome assemblies for *Buxus* and *Tetracentron*, along with proteomes for four publicly available eudicot genomes—*Arabidopsis thaliana, Aquilegia coerulea, Nelumbo nucifera,* and *Vitis vinifera* (Supplementary Data 8)—were provided as evidence. Custom repeat libraries for genome masking were produced according to the MAKER-P advanced protocol[54] using LTRharvest[55], LTRdigest[56], MITE-Hunter[57], RepeatModeler[58], and RepeatMasker[59]. Gene models were predicted from the masked assemblies using the SNAP[60] and Augustus[61] ab initio predictors after three rounds of training on interim high-quality (AED <= 0.25; length >= 50 amino acids) and BUSCO gene models, respectively.

**Phylogenetic analyses**. Three phylogenetic data sets were compiled from translated transcriptomes or genome-annotated proteomes for 40 angiosperms (Supplementary Data 8). Conserved single-copy land plant genes were identified by BUSCO[43] analyses with the embryophyta_odb10 data set, orthologs of the Angiosperms353 loci[44] were collected by BLAST searches seeded with *Amborella trichopoda* proteins, and orthogroups were circumscribed by Orthofinder[45]. For all data sets, protein sequences were aligned using MAFFT[62] and converted to codon alignments using PAL2NAL[63], which were refined in three successive rounds of sequence filtering and trimming using trimAl[64]. Initially, sequences with less than 50% residue overlap over >70% of their length were removed to discard any potentially spurious homologs. The passing sequences were next trimmed with trimAl's heuristic automatic method (-automated1) and filtered again as above to

remove sequences that might contribute extensive missing data to the phylogenetic matrix. Alignments with fewer than 4 sequences, and missing representatives of either Buxales or Trochodendrales, were discarded. After all filtering steps, 1248 BUSCOs, 346 Angiosperms353 loci, and 2573 orthogroups were retained for phylogenetic analyses. Maximum likelihood (ML) trees for the single-copy data sets were inferred from alignments of individual loci as well as concatenations of these, produced with FASconCAT[65], using RAxML[66] with the GTR + gamma model of nucleotide evolution and 1000 bootstrap replicates. Concatenated alignments were analyzed using a partition scheme that defines individual genes as units for parameter optimization. Partitioned Bayesian Inference analyses were run with MrBayes with the GTR + I + G model for all partitions. Two independent parallel runs of four Metropolis-coupled Monte Carlo Markov Chains were run for 10 million generations with sampling every 1000 generations. Majority rule consensus trees and posterior probabilities of bipartitions were computed after discarding the first 25% of the sampled trees as burn-in. Orthogroup trees were inferred with IQ-Tree[67] with the best substitution model selected from among those implemented in RAxML and 1000 ultrafast bootstrap replicates. ASTRAL-III[68] and ASTRAL-Pro[69] were used to infer the species trees from single- and multi-copy gene trees, respectively, under the multi-species coalescent. DensiTree[70] was used for visualizations of discordance among a subset of single-copy gene trees without missing taxa.

**Comparative genomics of polyploidy**. CoGe's SynMap and FractBias programs were used to perform genome alignments and fractionation bias calculations. FractBias analyses were conducted using all genes in the target genomes and syntenic depth settings in accordance with ploidy levels of respective genomes, as revealed by SynMap plots. All analyses can be regenerated on the CoGe platform (see Code availability below). For synteny-guided phylogenomic analyses, intergenomic alignments were produced and screened to identify all syntenic homologs (syntelogs) present in ratios of up to 3:2:2:2:4 in *Vitis, Aquilegia, Buxus, Nelumbo*, and *Tetracentron*, respectively, using MCscan[71]. This collection of syntenic homologs was divided into seven pools in accordance with the major synteny blocks conserved across these eudicot genomes (as identified by SynMap and FractBias mappings, and which correspond to ancestral eudicot chromosomes). Unique identifiers for individual loci were replaced by 'Species_chromosome' codes to create comparable phylogenetic matrices and trees for coalescence-based phylogenetic analyses as outlined above.

**Ancestral genomes**. To build the three ancestral genomes indicated in Fig. 4, we use the *RACCROCHE* pipeline[34]. Briefly, *RACCROCHE* uses all the syntenically validated homolog pairs generated by SynMap and builds disjoint gene families based on the principle that a gene homologous (orthologous or paralogous) with any gene in a family must also be a member of that family. For each genome, *RACCROCHE* extracts a set of 'generalized' adjacencies, namely all oriented pairs of genes within the same window containing seven consecutive genes. The pairs are represented by the non-adjacent ends of the two genes. The genes in these pairs are then labeled according to the gene families to which they belong. Each ancestor node has three incident branches, partitioning the tree into three subtrees defined by the one incoming edge (its ancestor) and two outgoing edges (its descendants). If an adjacency is found anywhere in any of the genomes in two or three of these subtrees, it is considered a candidate adjacency. With candidate adjacencies weighted as 2 or 3 according to the number of occurrences in subtrees, a maximum weight matching (MWM) of gene ends constructs the highest weight sets of compatible contiguous adjacencies (ancestral contigs). A gene end can only be matched to one end of another gene, so that these ancestral contigs are guaranteed to be linearly, or very occasionally circularly, ordered. Inversions with breakpoints within windows of seven consecutive genes will preserve common adjacencies between two genomes, but not reading directions within the window. Common adjacencies are our primary concern, so we do not use reading direction information in MWM. Circular contigs were linearized by breaking an adjacency of lowest weight. The ancestral contigs from MWM solutions were then aligned to chromosomes of modern genomes, and co-occurring contigs were clustered to assemble ancestral chromosomes. A complete-linkage clustering was applied to the correlations of contigs' co-occurrence to assemble ancestral chromosomes[72]. To aid in future studies of the genomic organization of gene function, a GO-term enrichment analysis of the members of each gene family was implemented to produce a functional annotation for the inferred ancestral genes. The functional annotations of ancestral genomes can be downloaded from https://git.cs.usask.ca/buxus/buxus-tetra.

**Reporting summary**. Further information on research design is available in the Nature Research Reporting Summary linked to this article.

## Data availability

All raw sequence reads used in this study have been deposited in NCBI under the BioProject accession numbers PRJNA549075, PRJNA547721, and PRJNA548936. In addition, the *Buxus* and *Tetracentron* genome assemblies, associated annotation files, and predicted CDS and protein sequences, along with all phylogenetic data sets analyzed here, and ancestral genome reconstructions have been deposited in the Dryad Digital

Repository [https://doi.org/10.5061/dryad.cjsxksn6d][73]. Source data are provided with this paper.

## Code availability

Custom scripts and command-line arguments have been deposited in GitHub [https://github.com/andrechanderbali/Buxus-Tetracentron-Genomes].

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

## Acknowledgements
This work was supported by National Science Foundation grants DEB-1455601 to A.S.C., DEB-1457440 to D.G.H., DEB-2030871 to V.A.A., and Discovery grants to L.J. and to D.S. from the Natural Sciences and Engineering Research Council of Canada. D.S. holds the Canada Research Chair in Mathematical Genomics. We thank Brent Berger, Ray Larson, and Veronica Di Stilio for aid in plant collection and DNA extraction and Hanqi Ye for bioinformatics discussion.

## Author contributions
A.S.C., D.E.S., D.G.H., D.S., P.S.S., and V.A.A. conceived and designed the study. A.S.C. generated the whole-genome and transcriptome assemblies, performed phylogenetic and comparative genomics analyses, and drafted the primary manuscript. D.S., L.J., and Q.X. generated and analyzed the ancestral genome reconstructions. Y.Z., E.C., and V.A.A. analyzed data. S.J. provided data. Additional text and discussion were provided by D.E.S., D.G.H., D.S., L.J., J.Z., P.S.S., and V.A.A. All authors approved the final manuscript.

## Competing interests
The authors declare no competing interests.
