## [Peer Review File · Nature Communications]

Buxus and Tetracentron genomes help resolve eudicot genome historyReviewers' Comments:

Reviewer #1:

Remarks to the Author:

Summary:

Ancient polyploidy events complicate the inference of the fascinating evolutionary history of flowering plants. Despite several efforts being made, our understanding of the diversification of the eudicot clade remains limited. The clade is divided into basal eudicots (Buxales, Trochodendrales, Ranunculales, Proteales) and core eudicots (Gunneridae). This work sheds light on the ancient polyploidy events that eventually led to the formation of the hexaploid genome (referred to as "gamma") and that is considered ancestral to all core eudicots. While evidence in earlier literature supports the hypothesis that two independent whole genome duplication (WGD) events were involved in the formation of gamma, their phylogenetic placement and their role in the diversification of eudicots remains unclear to this day. On the one side, it is debated whether the earlier WGD is shared by all eudicots or the core eudicots alone. On the other side, arguments have been made that the hexaploid genome has been a result of an inter-lineage hybridization of ancestral eudicot lineages. This work provides strong evidence that both WGD events are exclusive to core eudicots and rejects the hypothesis of inter-lineage hybridization. In doing so, it also resolves two competing hypotheses on the evolutionary relationships between core eudicots, Buxales, and Trochodendrales: one postulates that both, Buxales and Trochodendrales, are collectively sister groups of the core eudicots, whereas the other proposes that Buxales or Trochodendrales are individually sibling of the core eudicots while the other group branches off at an earlier point in time. The authors provide strong evidence for the latter, by constructing and analyzing high-resolution assemblies from long read sequencing data of *Buxus sinica* and *Tetracentron sinense*, two representatives of the mentioned orders. Although the exact phylogenetic placement and other details of the circumstances of the formation of gamma remain unresolved, this work brings us a substantial step closer towards that goal.

Strong points:

- This work makes an exciting and comprehensive effort towards understanding early eudicot evolution and I expect it to be well-received beyond the plant research community.
- The high-resolution assemblies of the basal eudicots are instrumental in settling several open questions about the evolution of the hexaploid ancestor of core eudicots and are valuable resources for the study of eudicot evolution.
- The gene order reconstructions of the three ancestral genomes is the first serious attempt of this kind for the ancestral genomes that constitute the stem lineage of eudicots. The presented protochromosomes will be yet another valuable resource for the field of research that studies the genome architecture of ancient plants.
- The manuscript is well written, I have only minor suggestions to improve the presentation of this work.

Weak points:

- The manuscript is vague about the data that has been collected and analyzed apart from the PacBio sequencing data of *Buxus sinica* and *Tetracentron sinense*. While the data section points to an additional repository of "multispecies RNA-seq data" and Supplemental Table 8 lists 9 additional species for which data was collected in this work, I haven't seen the data being mentioned in the Methods or the Results and Discussion sections. It is not clear where the RNA-seq data was used in this study, what other data from the 9 species was obtained, and how it was processed.
- The resolution of Figures 1 and 2 is in parts so low that they cannot be deciphered. This impacted my understanding of the presented results and my ability to scrutinize related parts of the manuscript,

in particular those related to Figure 2c, where I think these plots carry crucial information. But I fault less the authors than the editorial staff of this journal who should check for such technicalities prior to relaying the manuscript to the reviewers.

Suggestions:

- Please add the missing information about of **all** data that has been collected and analyzed to the manuscript. Also, if the supplementary table would detail what kind of data was collected, it will ease the navigation to the data for avid readers.
- Please improve the resolution all figures, in particular that of Figures 1 and 2.
- Figure 3, Supplemental Table 5 and various places in the text: I find it confusing to refer to the 7 ancestral chromosomes as "Ancestor 1", "Ancestor 2", etc., because the text also refers to "Ancestor 1", "Ancestor 2" and "Ancestor 3" in their collectivity as the three ancestral genomes of the eudicot stem lineage. Please improve your notation here!
- The section "data availability" mentions a comprehensive set of data that will be made available, but I don't see the reconstructed protochromosomes of three ancestral genomes listed there. It would be nice to see those being released to the public, too.
- Figure 1 and 2: caption stops prematurely in an unfinished sentence.
- A consistent coloring scheme of ancestral chromosomes in Figures 3 and 4 Supplemental Figure 4 would substantially ease the orientation for the readers
- Page 5, Lines 203ff: the term "syntenic depth" should be explained.
- Page 5, 2nd paragraph (Lines 238ff) - this paragraph contains some redundant sentences, such as the summarizing sentences starting in Line 438, that can be omitted.
- Page 6, Line 248: "lineages Altogether" - period missing
- Page 10, line 434: "Because the left end of a gene can only be matched to the right end of another gene, these ancestral contigs are guaranteed to be linear." What do you mean by "linear" here? If your notion does not include circular chromosomes, this statement does not hold true in general. Otherwise, please make the statement more precise.
- Supplemental Figure 3 should have a caption that describes the shown plots. Also, the resolution was too low for me to comprehend the figure.

Reviewer #2:

Remarks to the Author:

In this manuscript, the authors present the first genome assemblies for *Buxus* and *Tetracentron*, two key species that are ideally suited to investigate the evolution of eudicot genomes, including early polyploid events (e.g. *At-gamma*). These two new genome assemblies complement those already publicly available for other early-diverging eudicot lineages. The scaffold N50s are excellent for these two genomes, but the reported BUSCO scores are quite low, much lower than what I was expecting based on the assembly metrics. However, the overall quality of the genome is more than sufficient for the questions being targeted by the authors in the manuscript.

They obtained new and improved estimates for relationships within eudicots (e.g. *Buxales/Trochodendrales* clade being sister to core eudicots) and the placement and contributors of the *At-gamma* event. Furthermore, they utilized a new tool (RACCROCHE) to infer the ancestral genomes at important eudicot nodes. This should serve as an additional useful resource for the community. Overall, they present a robust dataset from multiple sources that collectively are inconsistent with previous proposed model for the *At-gamma* event. These findings should be of general interest to plant evolutionary biology community.

Just a few comments:

1. Figure 4, the right edge is cutoff. It was sufficient to review.
2. The genome and annotation files should be made available in the same Dryad directory as the phylogenomic data. These data needs to be provided in an additional public data server besides just CoGe.
3. The CoGe URLs provided in the manuscript (page 10) were unable to be reviewed. The data was set to private. Not an issue. But these need to be made available before the manuscript is published.

Reviewer #3:

Remarks to the Author:

I have enjoyed reading the manuscript titled "Buxus and Tetracentron genomes, ancient polyploidy, and the search for gamma" by Chanderbali and collaborators. The manuscript presents the genomes of two early-diverging eudicot plants, which are used in combination with other plant genomes to test the presence of a Whole Genome Duplication (WGD) in the origin of eudicots. They analyse these genomes using phylogenomics, synteny analyses, and reconstructing the genome of the first eudicot at the chromosome level. I want to stress how interesting and amazing is that the authors manage to reconstruct the ancestral genome of eudicots at the chromosome level. On the other side, not much is said about the genes (or their functions) involved in these WGD or the ancestral genome. Overall, the analyses are robust and adhere to the standards of the field, and the article is well-written and presented. I'd like to congratulate the authors for their work and their efforts.

Before I share my thoughts on the paper, I must clarify that, while I have experience in comparative genomics, phylogenomics, and have work on early land plants, I am not an expert on WGDs within tracheophytes or the fine phylogenetics of eudicots. I apologise if I miss any minutiae about these topics, or make a poor suggestion out of ignorance. However, I hope this makes me representative of the average reader of this type of paper in Nature Comms, and anything I may misunderstand may provide an opportunity for clarification in the text. As I said above, I think both analyses and the manuscript are great, but I have some comments and questions that I hope the authors can kindly address.

If I could only give advice on one topic, it would be about the phylogenomic analyses. First, I think branch lengths are important to assess trees (Figure 2a) and identify long branched taxa, even more with multimarker supermatrices that tend to inflate statistical supports. Second, the phylogenetic analyses section states the use of proteins from BUSCO in the alignments and trimming, but the analyses with RAxML seem to be done with a nucleotide model, I was wondering what is the rationale behind this choice. Third, based on the "Phylogenetic analyses" section, the concatenated tree was run only with 100 bootstrap replicates, which is insufficient from a statistical point of view. This is not a big dataset, and 1000 replicates should be doable with IQTree.

There is no indication of a partitioning scheme being used in the concatenated analysis, which I think is a must with these multimarker datasets, and may help with the discordance between concatenation and coalescence. Similarly, one would expect Bayesian Inference trees for this type of dataset. One suggestion to explore this discrepancy is the use of closer outgroups. For example, a complementary analysis could be to run a tree using the sister group to Buxus + Trocho + Core eudicots as an outgroup. Another idea would be to run the analyses with ALE (Amalgamated Likelihood Estimation), which should be easy as the input files are the same trees used in ASTRAL.

Another important point is data availability. The manuscript states that the raw reads will be available in SRA. However, raw reads are not conducive to follow up studies or post-publication peer review. Assemblies, protein models, and annotations should be made available, both for reproducibility purposes and to facilitates analyses by other researchers, avoiding any "available upon request"

clause.

I might be biased here, and I know articles have strong space limits, but I was expecting some information or analyses about the genes involved in the WGD (e.g., the ones in Figure 4b), their retention postWGD, their biological functions, etc. As much as I think this paper provides strong support for the nature of the gamma WGD, and the reconstruction of the ancestral eudicot genome at chromosome level is awesome, I would hope for some link between chromosomal-level events and gene biology, which may tell us more about the evolutionary context of the origin of eudicots.

Finally, I am not sure if this might be caused by the submissions system, but I couldn't read some of the figures. I am not that old! I read the paper on a computer, and no matter how much I zoomed in, I couldn't read the text within Figures 1 or 2c, they are just anamorphous mass of pixels. I couldn't find any link to the original full-size figures.

Some minor suggestions:

1) I found the title poetic but not very attractive. Most importantly, it is not informative unless you are familiar with these plant genera or the naming of WGDs within plants. I didn't know these organisms and, while I am aware of the WGDs within plants, it took me a bit to link the 'gamma' with the WGDs as I was not even sure this was about plants. Given the audience of Nature Comms, I'd suggest a title that makes clear what is the biological question (e.g., WGD in eudicots), and hints at the main findings.

2) Linked to the previous point, the manuscript is not accessible to non-plant experts. And even for plant experts, some context may help better to understand the paper. For example, lines 50 to 66 are probably the most important lines in the introduction to understand in this paper, but they are very hard to follow. Some more obscure concepts could be explained better (e.g., fractionation, syntelogs, palaeologs, X:Y syntenic depth), and importantly, the difference between the 2 competing hypotheses could be more elaborate.

3) Figure 3b, I failed to find an explanation of what the "Ancestors" are, apologies if I missed it. I understand each tree comes from all the genes inferred to be together in one ancestral chromosome, like in Figure 4b?

4) Lines 257 and 259, sentence mentions a hypothesis in which one of the third gamma genomes was contributed by a now-extinct species. Is this not the case for all three donor genomes? Unless the species we believe that were sequenced today were around 160 Mya. I guess the text meant a lineage that left no descendants?

REVIEWER COMMENTS

Reviewer #1 (Remarks to the Author):

Summary:

Ancient polyploidy events complicate the inference of the fascinating evolutionary history of flowering plants. Despite several efforts being made, our understanding of the diversification of the eudicot clade remains limited. The clade is divided into basal eudicots (Buxales, Trochodendrales, Ranunculales, Proteales) and core eudicots (Gunneridae). This work sheds light on the ancient polyploidy events that eventually led to the formation of the hexaploid genome (referred to as "gamma") and that is considered ancestral to all core eudicots. While evidence in earlier literature supports the hypothesis that two independent whole genome duplication (WGD) events were involved in the formation of gamma, their phylogenetic placement and their role in the diversification of eudicots remains unclear to this day. On the one side, it is debated whether the earlier WGD is shared by all eudicots or the core eudicots alone. On the other side, arguments have been made that the hexaploid genome has been a result of an inter-lineage hybridization of ancestral eudicot lineages. This work provides strong evidence that both WGD events are exclusive to core eudicots and rejects the hypothesis of inter-lineage hybridization. In doing so, it also resolves two competing hypotheses on the evolutionary relationships between core eudicots, Buxales, and Trochodendrales: one postulates that both, Buxales and Trochodendrales, are collectively sister groups of the core eudicots, whereas the other proposes that Buxales or Trochodendrales are individually sibling of the core eudicots while the other group branches off at an earlier point in time. The authors provide strong evidence for the latter, by constructing and analyzing high-resolution assemblies from long read sequencing data of *Buxus sinica* and *Tetracentron sinense*, two representatives of the mentioned orders. Although the exact phylogenetic placement and other details of the circumstances of the formation of gamma remain unresolved, this work brings us a substantial step closer towards that goal.

Strong points:

- This work makes an exciting and comprehensive effort towards understanding early eudicot evolution and I expect it to be well-received beyond the plant research community.
- The high-resolution assemblies of the basal eudicots are instrumental in settling several open questions about the evolution of the hexaploid ancestor of core eudicots and are valuable resources for the study of eudicot evolution.
- The gene order reconstructions of the three ancestral genomes is the first serious attempt of this kind for the ancestral genomes that constitute the stem lineage of eudicots. The presented protochromosomes will be yet another valuable resource for the field of research that studies the genome architecture of ancient plants.
- The manuscript is well written, I have only minor suggestions to improve the presentation of this work.

Weak points:

- The manuscript is vague about the data that has been collected and analyzed apart from the PacBio sequencing data of *Buxus sinica* and *Tetracentron sinense*. While the data section points to an additional repository of "multispecies RNA-seq data" and Supplemental Table 8 lists 9 additional species for which data was collected in this work, I haven't seen the data being mentioned in the Methods or the Results and Discussion sections. It is not clear where the RNA-seq data was used in this study, what other data from the 9 species was obtained, and how it was processed.

- The resolution of Figures 1 and 2 is in parts so low that they cannot be deciphered. This impacted my understanding of the presented results and my ability to scrutinize related parts of the manuscript, in particular those related to Figure 2c, where I think these plots carry crucial information. But I fault less the authors than the editorial staff of this journal who should check for such technicalities prior to relaying the manuscript to the reviewers.

Suggestions:

- Please add the missing information about of *all* data that has been collected and analyzed to the manuscript. Also, if the supplementary table would detail what kind of data was collected, it will ease the navigation to the data for avid readers.

Thank you for pointing out this oversight. We have added fields to Supplemental Table 8 listing the type of data analyzed (Genome Sequence, RNA-seq) and the topics of analysis (Eudicot Phylogeny, Sub-genome Phylogenomics, Ancestral Genome Reconstruction). We have also added text under the Methods section describing our rationale for and processing of RNA-seq data, as follows.

RNA-seq Data

Transcriptome assemblies were produced for *Buxus sinica* and *Tetracentron sinense* to aid annotation of their genome assemblies. We also produced transcriptome assemblies for six additional early-diverging eudicots (*Buxus sempervirens*, *Meliosma dillenifolia*, *Nelumbo lutea*, *Sabia emarginata*, *Sabia swinhonei*, *Trochodendron aralioides*), as well as the core eudicot (*Gunnera manicata*), to improve taxon sampling in phylogenetic analyses. Paired-end RNA-seq libraries were constructed from polyA selected total RNA extracted from floral and/or leaf tissues (Supplemental Table 8), and sequenced using the Illumina HiSeq 3000 system. Reads were trimmed with Trimmomatic v. 0.39⁵² and assembled using Trinity v.2.12.0⁵³. Coding DNA (CDS) and protein sequences were predicted with TransDecoder v.5.0.2 (<http://transdecoder.github.io>).

- Please improve the resolution all figures, in particular that of Figures 1 and 2.

We apologize for the figure resolution issues. High quality versions of all figures have been produced for the revised submission.

- Figure 3, Supplemental Table 5 and various places in the text: I find it confusing to refer to the 7 ancestral chromosomes as "Ancestor 1", "Ancestor 2", etc., because the text also refers to "Ancestor 1",

"Ancestor 2" and "Ancestor 3" in their collectivity as the three ancestral genomes of the eudicot stem lineage. Please improve your notation here!

Thanks for pointing out this source of confusion. We have changed the notation of the ancestral chromosomes in Figure 3, Supplemental Table 5 and the text to "Ancestral Chromosome 1", "Ancestral Chromosome 2", "Ancestral Chromosome 3", etc.

- The section "data availability" mentions a comprehensive set of data that will be made available, but I don't see the reconstructed protochromosomes of three ancestral genomes listed there. It would be nice to see those being released to the public, too.

Thank you for this suggestion. We have posted our reconstructed protochromosomes to a publicly accessible git repository at <https://git.cs.usask.ca/buxus/buxus-tetra>, and to our Dryad submission.

- Figure 1 and 2: caption stops prematurely in an unfinished sentence.

Figure captions are provided as a separate section in the resubmitted manuscript. We anticipate this formatting change will remedy any inadvertent loss of text.

- A consistent coloring scheme of ancestral chromosomes in Figures 3 and 4 Supplemental Figure 4 would substantially ease the orientation for the readers.

We agree that a consistent color scheme would be ideal, and have modified the colors of ancestral chromosomes in Fig 3 to match those in Figure 4 and Supplemental Figure 4 (now Supplemental Figure 7).

- Page 5, Lines 203ff: the term "syntenic depth" should be explained.

We have revised the paragraph in which this term is used as follows:

Page 6, Line 159: Pairwise analyses of intergenomic collinearity (macrosynteny) and fractionation patterns identify extensive regions of early-diverging eudicot genomes shared with the *gamma*-derived hexaploid genome of *Vitis*, and each other (Supplemental Figures 2 and 3). The ratios of syntenic depths (the number of times a genomic region is syntenic to regions in another genome) in these comparisons reflect the number of subgenomes, or level of ploidy, for the respective species. Thus, we see 2:3 syntenic depth between *Buxus* and *Vitis*, and 4:3 syntenic depth between *Tetracentron* and *Vitis*, while *Tetracentron* to *Buxus* is 4:2 in syntenic depth. Likewise, as previously reported, *Aquilegia* and *Nelumbo* each exhibit 2:3 syntenic depth with *Vitis*, and 2:2 with each other.

- Page 5, 2nd paragraph (Lines 238ff) - this paragraph contains some redundant sentences, such as the summarizing sentences starting in Line 438, that can be omitted.

We are unsure about which summarizing sentences the reviewer is referring. Lines 238 (Results) and 438 (Methods) are in different manuscript sections, and neither offers summarizing sentences.

- Page 6, Line 248: "lineages Altogether" - period missing

The missing period has been added.

- Page 10, line 434: "Because the left end of a gene can only be matched to the right end of another gene, these ancestral contigs are guaranteed to be linear." What do you mean by "linear" here? If your notion does not include circular chromosomes, this statement does not hold true in general. Otherwise, please make the statement more precise.

Thank you for pointing out this potential point of confusion. We did not mean to refer to circular chromosomes. The "linear" here refers to the upstream/downstream ordering on reconstructed protochromosomes. To make this more clear, we have changed the text as follows:

Page 13, Line 378: "...these ancestral contigs are guaranteed to be linearly ordered in upstream/downstream orientation".

- Supplemental Figure 3 should have a caption that describes the shown plots. Also, the resolution was too low for me to comprehend the figure.

We apologize for the low figure resolution and brief caption. High quality versions of all figures have been produced for the revised submission.

We have provided the following caption to Supplemental Figure 3 (now Supplemental Figure 6):
"FractBias plots comparing *Vitis vinifera* (target genome) with *Aquilegia coerulea*, *Buxus sinica*, *Nelumbo nucifera* and *Tetracentron sinense* (query genomes). Each item in the figure (19 rows x 4 columns) depicts a pair-wise comparison between one *Vitis* chromosome (1 through 19) and the corresponding syntenic regions of the four target genomes. X-axes correspond to iterated sliding windows of 100 genes on the target chromosome. Y-axes indicate gene retention percentages at syntenic locations on query chromosomes.

Reviewer #2 (Remarks to the Author):

In this manuscript, the authors present the first genome assemblies for *Buxus* and *Tetracentron*, two key species that are ideally suited to investigate the evolution of eudicot genomes, including early polyploid events (e.g. At-gamma). These two new genome assemblies complement those already publicly available for other early-diverging eudicot lineages. The scaffold N50s are excellent for these two genomes, but the reported BUSCO scores are quite low, much lower than what I was expecting based on the assembly metrics. However, the overall quality of the genome is more than sufficient for the questions being targeted by the authors in the manuscript.

They obtained new and improved estimates for relationships within eudicots (e.g. Buxales/Trochodendrales clade being sister to core eudicots) and the placement and contributors of the At-gamma event. Furthermore, they utilized a new tool (RACCROCHE) to infer the ancestral genomes at important eudicot nodes. This should serve as an additional useful resource for the community. Overall,

they present a robust dataset from multiple sources that collectively are inconsistent with previous proposed model for the At-gamma event. These findings should be of general interest to plant evolutionary biology community.

Just a few comments:

1. Figure 4, the right edge is cutoff. It was sufficient to review.
2. The genome and annotation files should be made available in the same Dryad directory as the phylogenomic data. These data needs to be provided in an additional public data server besides just CoGe.

We agree with these comments. We have included these files in the Dryad Digital Repository, which may be accessed during the review period at https://datadryad.org/stash/share/SIQPNI2Io1N3i-7C9j56lnBIZAwq3M1oQQcJdS0s_4U. We have modified our “Data Availability” section accordingly, as follows:

Page 13, Line 398: All of the raw sequence reads used in this study have been deposited in NCBI under the BioProject accession numbers PRJNA549075 (*Buxus sinica*), PRJNA547721 (*Tetracentron sinense*), and PRJNA548936 (multispecies RNA-seq data). Additionally, the *Buxus* and *Tetracentron* genome assemblies, associated annotation files, and predicted CDS and protein sequences, along with all phylogenetic data sets analyzed here, and ancestral genome reconstructions have been deposited in the Dryad Digital Repository (<https://doi.org/10.5061/dryad.cjsxksn6d>).

3. The CoGe URLs provided in the manuscript (page 10) were unable to be reviewed. The data was set to private. Not an issue. But these need to be made available before the manuscript is published.

We agree. The CoGe URLs will be made available to the public once the manuscript is accepted for publication.

Reviewer #3 (Remarks to the Author):

I have enjoyed reading the manuscript titled “*Buxus* and *Tetracentron* genomes, ancient polyploidy, and the search for gamma” by Chanderbali and collaborators. The manuscript presents the genomes of two early-diverging eudicot plants, which are used in combination with other plant genomes to test the presence of a Whole Genome Duplication (WGD) in the origin of eudicots. They analyse these genomes using phylogenomics, synteny analyses, and reconstructing the genome of the first eudicot at the chromosome level. I want to stress how interesting and amazing is that the authors manage to reconstruct the ancestral genome of eudicots at the chromosome level. On the other side, not much is said about the genes (or their functions) involved in these WGD or the ancestral genome. Overall, the analyses are robust and adhere to the standards of the field, and the article is well-written and presented. I’d like to congratulate the authors for their work and their efforts.

Before I share my thoughts on the paper, I must clarify that, while I have experience in comparative genomics, phylogenomics, and have work on early land plants, I am not an expert on WGDs within tracheophytes or the fine phylogenetics of eudicots. I apologise if I miss any minutiae about these topics, or make a poor suggestion out of ignorance. However, I hope this makes me representative of the average reader of this type of paper in Nature Comms, and anything I may misunderstand may provide an opportunity for clarification in the text. As I said above, I think both analyses and the manuscript are great, but I have some comments and questions that I hope the authors can kindly address.

If I could only give advice on one topic, it would be about the phylogenomic analyses. First, I think branch lengths are important to assess trees (Figure 2a) and identify long branched taxa, even more with multimarker supermatrices that tend to inflate statistical supports.

Thank you for this suggestion. We agree that branch lengths provide important information, and have replaced the cladograms in Fig 2a with phylograms showing relative branch lengths. Newly provided phylogenetic trees in Supplemental Figures 2-4 are also shown as phylograms.

Second, the phylogenetic analyses section states the use of proteins from BUSCO in the alignments and trimming, but the analyses with RAxML seem to be done with a nucleotide model, I was wondering what is the rationale behind this choice.

Perhaps the reviewer misunderstood our methods. Our methods stated “For all data sets, protein sequences were aligned using MAFFT v.7.402⁵⁸ and converted to codon alignments using PAL2NAL v.14⁵⁹, which were refined in three successive rounds of sequence filtering and trimming using trimAl⁶⁰. “

The alignments of protein sequences were converted to nucleotide (codon) alignments to leverage the added phylogenetic signal provided by nucleotides over amino acids.

Third, based on the “Phylogenetic analyses” section, the concatenated tree was run only with 100 bootstrap replicates, which is insufficient from a statistical point of view. This is not a big dataset, and 1000 replicates should be doable with IQTree.

On double-checking our analysis logs, we did conduct 1000 bootstrap replicates for the concatenated tree analysis. These analyses were done with RAxML. We have corrected this typographical error in the manuscript as follows:

Page 11, Line 333: Maximum likelihood (ML) trees for the single-copy data sets were inferred from alignments of individual loci as well as concatenations of these, produced with FASconCAT⁶¹, using RAxML v.8.2.12⁶² with the GTR+gamma model of nucleotide evolution and 1000 bootstrap replicates.

There is no indication of a partitioning scheme being used in the concatenated analysis, which I think is a must with these multimarker datasets, and may help with the discordance between concatenation and coalescence.

We apologize for this oversight. We did use partition schemes for the concatenated datasets, and have added the following text to the manuscript to reflect this strategy.

Page 11, Line 337: “Concatenated alignments were analyzed using a partition scheme that defines individual genes as units for parameter optimization.”

Similarly, one would expect Bayesian Inference trees for this type of dataset.

Thanks for the suggestion to provide Bayesian Inference analyses of our multimarker phylogenetic data sets. We have used MrBayes to conduct partitioned BI analyses on the Angiosperms353 and BUSCO supermatrices, and included the respective consensus trees as Supplemental Figure 4. We have modified the relevant manuscript sections as follows:

Page 5, Line 133: Concatenated data sets of the SCN loci, whether analyzed in Maximum Likelihood (Fig 2a, right panel, Supplemental Figure 2b) or Bayesian Inference (Supplemental Figure 4) frameworks, recover Buxales alone as the core eudicot sister group, with Trochodendrales as sister to this Buxales+core eudicot clade.

Page 11, Line 338: Partitioned Bayesian Inference analyses were run with MrBayes v.3.2.6 with the GTR+I+G model for all partitions. Two independent parallel runs of four Metropolis-coupled Monte Carlo Markov Chains were run for 10 million generations with sampling every 1000 generations. Majority rule consensus trees and posterior probabilities of bipartitions were computed after discarding the first 25% of the sampled trees as burn-in.

One suggestion to explore this discrepancy is the use of closer outgroups. For example, a complementary analysis could be to run a tree using the sister group to Buxus + Trocho + Core eudicots as an outgroup.

We are not entirely sure what outgroup strategy the reviewer is suggesting. The sister group of the Buxus + Trocho + Core eudicots is included in the analysis. We therefore feel we have handled these analyses in the appropriate fashion.

Another idea would be to run the analyses with ALE (Amalgamated Likelihood Estimation), which should be easy as the input files are the same trees used in ASTRAL.

Thanks for the suggestion. Our understanding is that the Amalgamated Likelihood Estimation approach, as stated at the GitHub repository (<https://github.com/ssolo/ALE>), “reconciles a sample of gene trees with a species tree”. Since our goal is to produce a species tree, we suspect ALE is not suitable for our purposes.

Another important point is data availability. The manuscript states that the raw reads will be available in SRA. However, raw reads are not conducive to follow up studies or post-publication peer review. Assemblies, protein models, and annotations should be made available, both for reproducibility purposes and to facilitate analyses by other researchers, avoiding any “available upon request” clause.

We agree with these comments, and as suggested also by another reviewer (please see above), we have deposited the genome assemblies, annotation files, and corresponding CDS and protein models in the Dryad Digital Repository, which may be accessed during the review period at https://datadryad.org/stash/share/SIQPNI2Io1N3i-7C9j56lnBIZAwq3M1oQQcJdS0s_4U.

I might be biased here, and I know articles have strong space limits, but I was expecting some information or analyses about the genes involved in the WGD (e.g., the ones in Figure 4b), their retention postWGD, their biological functions, etc. As much as I think this paper provides strong support for the nature of the gamma WGD, and the reconstruction of the ancestral eudicot genome at chromosome level is awesome, I would hope for some link between chromosomal-level events and gene biology, which may tell us more about the evolutionary context of the origin of eudicots.

We agree that the functional consequences of the gamma WGDs are important to investigate, but these are really beyond the scope of a single paper. We prefer, therefore, to assess these in future analyses. Meanwhile, to facilitate such effort by interested researchers, the GO annotations for genes placed on ancestral genomes have added to our Dryad submission and also made public at <https://git.cs.usask.ca/buxus/buxus-tetra>. These links are provided in the manuscript.

Finally, I am not sure if this might be caused by the submissions system, but I couldn't read some of the figures. I am not that old! I read the paper on a computer, and no matter how much I zoomed in, I couldn't read the text within Figures 1 or 2c, they are just anamorphous mass of pixels. I couldn't find any link to the original full-size figures.

We apologize for this inconvenience. High quality versions of all figures have been supplied to the editors with the revised submission.

Some minor suggestions:

1) I found the title poetic but not very attractive. Most importantly, it is not informative unless you are familiar with these plant genera or the naming of WGDs within plants. I didn't know these organisms and, while I am aware of the WGDs within plants, it took me a bit to link the 'gamma' with the WGDs as I was not even sure this was about plants. Given the audience of Nature Comms, I'd suggest a title that makes clear what is the biological question (e.g., WGD in eudicots), and hints at the main findings.

We welcome the reviewers' suggestions towards a more informative title for the non-plant reader. We have tried to incorporate these elements in a revised title that complies with journal length restrictions:

Original title: ***Buxus and Tetracentron genomes, ancient polyploidy, and the search for gamma***

Revised title: ***Buxus* and *Tetracentron* genomes help resolve eudicot phylogeny, ancient polyploidy, and paleogenomes**

2) Linked to the previous point, the manuscript is not accessible to non-plant experts. And even for plant experts, some context may help better to understand the paper. For example, lines 50 to 66 are probably the most important lines in the introduction to understand in this paper, but they are very hard to follow. Some more obscure concepts could be explained better (e.g., fractionation, syntelogs, palaeologs, X:Y syntenic depth), and importantly, the difference between the 2 competing hypotheses could be more elaborate.

We have revised the text where these terms are first used to add explanations:

Page 2, Line 57: ... extended period of random fractionation (loss of either copy of duplicated genomic regions following WGD) in the tetraploid intermediate

Page 4, Line 122: ... syntenic paralogs (paleologs)

Page 6, Line 159: Pairwise analyses of intergenomic collinearity (macrosyteny) and fractionation patterns identify extensive regions of early-diverging eudicot genomes shared with the *gamma*-derived hexaploid genome of *Vitis*, and each other (Supplemental Figures 5 and 6). The ratios of syntenic depths (the number of times a genomic region is syntenic to regions in another genome) in these comparisons reflect the number of subgenomes, or level of ploidy, for the respective species. Thus, we see 2:3 syntenic depth between *Buxus* and *Vitis*, and 4:3 syntenic depth between *Tetracentron* and *Vitis*, while *Tetracentron* to *Buxus* is 4:2 in syntenic depth. Likewise, as previously reported, *Aquilegia* and *Nelumbo* each exhibit 2:3 syntenic depth with *Vitis*, and 2:2 with each other.

Page 6, Line 174: Microsyteny (gene level) alignments within these major syteny blocks comprise 235 homologous loci and a total of 1837 syntelogs (genes derived from the same ancestral genomic region) useful for inferring the evolutionary history of ancestral chromosome 4 (see Supplemental Table 5 for the modern distribution and representation of each ancestral chromosome).

3) Figure 3b, I failed to find an explanation of what the “Ancestors” are, apologies if I missed it. I understand each tree comes from all the genes inferred to be together in one ancestral chromosome, like in Figure 4b?

The reviewer’s understanding is correct. A similar comment was made by Reviewer 1. To clarify this, we have changed the “Ancestor” labels in Figure 3b, Supplemental Table 5 and the text to “Ancestral Chromosome 1”, “Ancestral Chromosome 2”, “Ancestral Chromosome 3”, etc.

4) Lines 257 and 259, sentence mentions a hypothesis in which one of the third gamma genomes was contributed by a now-extinct species. Is this not the case for all three donor genomes? Unless the

species we believe that were sequenced today were around 160 Mya. I guess the text meant a lineage that left no descendants?

The reviewer's understanding is correct. We have replaced the phrase "extinct species" with "extinct lineage" to clarify this point as follows:

Page 7, Line 207: "As such, if hexaploidy was attained via a two-step process of sequential WGDs, the third of the *gamma* genomes must have been donated from a now extinct lineage that branched off the core eudicot ancestral line before the first tetraploidy event (Fig. 3c)."

Reviewers' Comments:

Reviewer #1:

Remarks to the Author:

The authors did an excellent job in revising the manuscript. I am very pleased with all changes, except one sentence that keeps bugging me:

"Because the left end of a gene can only be matched to the right end of another gene, these ancestral contigs are guaranteed to be linearly ordered in upstream/downstream orientation."

I think I understand now the described situation. The author's additional clause that "ancestral contigs are guaranteed to be linearly ordered in upstream/downstream orientation" makes me believe that these ancestral genes are partially ordered from left to right, so that a gene extremity can only be matched to a gene extremity on its right. If that is the case, I'd kindly ask the authors to revise the sentence accordingly. Referring to the "left" an "right" end of a gene is ambiguous, because genes themselves have an orientation defined by their reading direction. It should be made clear that the system of reference is not the reading direction of genes, but the partial ordering of the genes within the to-be-inferred ancestral contigs. I've attached a visualization of the alternative meaning of this ambiguous sentence.

Reviewer #2:

Remarks to the Author:

The authors have adequately addressed all of my previous concerns and comments.

Reviewer #3:

Remarks to the Author:

I want to thank the authors for their efforts in improving the paper. The text is more accessible now, and the additional analyses are welcome and make their results more robust.

I am probably to blame for a couple of misunderstandings in our last exchange. For example, when I suggested using different outgroups, I meant it as an experimental design. Outgroups and sister groups are not the same, the first is decided by the researchers for rooting the tree later (i.e., Amborella in this study), the second is a consequence of the topology (e.g., Buxus + Trocho are sister to Core eudicots). However, too distant outgroups can exacerbate LBA issues. Thus, I suggested running a tree using the sister group to Buxus + Trocho + Core eudicots (the clade including Macadamia and Sabia) as an outgroup, excluding all the other clades (comprising Papaver, Liriodendron, Oryza, and Amborella). Using a closer outgroup as an experimental design could minimise the concerns about LBA, but Buxus and Trochodendron don't seem to have long branches (one couldn't tell in the previous version of the manuscript).

Similarly, I agree with the description of ALE that the authors make in their rebuttal letter. But if one digs deeper in ALE documentation, the patterns of gene duplications/losses/transfers can be used to infer a species tree. This has been done for example to root the Tree of Life and find LUCA (check papers by Tom Williams). This is straightforward if one has the gene trees for all the genes, which you should have as this is the input for your ASTRAL analyses.

I'd suggest that the authors make available as supp data the output files from RAxML. Having access to the bootstrap trees and the partition schemes from RAxML may be useful to other researchers willing to reanalyse datasets.

I appreciate all the other changes and the lack of space (scope on the biology of the genes involved in the WGD). I also thank the authors for improving the resolution of the figures (much better!).

REVIEWER COMMENTS

Reviewer #1 (Remarks to the Author):

The authors did an excellent job in revising the manuscript. I am very pleased with all changes, except one sentence that keeps bugging me:

"Because the left end of a gene can only be matched to the right end of another gene, these ancestral contigs are guaranteed to be linearly ordered in upstream/downstream orientation."

I think I understand now the described situation. The author's additional clause that "ancestral contigs are guaranteed to be linearly ordered in upstream/downstream orientation" makes me believe that these ancestral genes are partially ordered from left to right, so that a gene extremity can only be matched to a gene extremity on its right. If that is the case, I'd kindly ask the authors to revise the sentence accordingly. Referring to the "left" an "right" end of a gene is ambiguous, because genes themselves have an orientation defined by their reading direction. It should be made clear that the system of reference is not the reading direction of genes, but the partial ordering of the genes within the to-be-inferred ancestral contigs. I've attached a visualization of the alternative meaning of this ambiguous sentence.

Thanks for the detailed explanation of the ambiguity introduced by our earlier revision. To address this issue more carefully, we have expanded our explanation of ancestral contig construction. The revised passage reads as follows:

"A gene end can only be matched to one end of another gene, so that these ancestral contigs are guaranteed to be linearly, or very occasionally circularly, ordered. Inversions with breakpoints within windows of seven consecutive genes will preserve common adjacencies between two genomes, but not reading directions within the window. Common adjacencies are our primary concern, so we do not use reading direction information in MWM. Circular contigs were linearized by breaking an adjacency of lowest weight."

Reviewer #2 (Remarks to the Author):

The authors have adequately addressed all of my previous concerns and comments.

Thanks for accepting our responses to your review.

Reviewer #3 (Remarks to the Author):

I want to thank the authors for their efforts in improving the paper. The text is more accessible now, and the additional analyses are welcome and make their results more robust.

I am probably to blame for a couple of misunderstandings in our last exchange. For example, when I suggested using different outgroups, I meant it as an experimental design. Outgroups and sister groups are not the same, the first is decided by the researchers for rooting the tree later (i.e., Amborella in this study), the second is a consequence of the topology (e.g., Buxus + Trocho are sister to Core eudicots).

However, too distant outgroups can exacerbate LBA issues. Thus, I suggested running a tree using the sister group to Buxus + Trocho + Core eudicots (the clade including Macadamia and Sabia) as an outgroup, excluding all the other clades (comprising Papaver, Liriodendron, Oryza, and Amborella). Using a closer outgroup as an experimental design could minimise the concerns about LBA, but Buxus and Trochodendron don't seem to have long branches (one couldn't tell in the previous version of the manuscript).

Similarly, I agree with the description of ALE that the authors make in their rebuttal letter. But if one digs deeper in ALE documentation, the patterns of gene duplications/losses/transfers can be used to infer a species tree. This has been done for example to root the Tree of Life and find LUCA (check papers by Tom Williams). This is straightforward if one has the gene trees for all the genes, which you should have as this is the input for your ASTRAL analyses.

I'd suggest that the authors make available as supp data the output files from RAxML. Having access to the bootstrap trees and the partition schemes from RAxML may be useful to other researchers willing to reanalyse datasets.

I appreciate all the other changes and the lack of space (scope on the biology of the genes involved in the WGD). I also thank the authors for improving the resolution of the figures (much better!).

Thanks for agreeing with our responses and for pointing us to the ALE literature. From these, we gather that ALE can be used to evaluate the likelihood of alternative species trees in the context of gene duplications/transfers/losses (DTL) patterns. For example, the alternative rooting positions for the Tree of Life evaluated in Williams et al. 2017 (<https://doi.org/10.1073/pnas.1618463114>). Likewise, we could conceivably evaluate alternative positions of Buxales and Trochodendrales relative to core eudicots using this approach. However, we are not experienced users of ALE and would prefer to conduct such analyses in collaboration with expert ALE authors (as in Williams et al. 2017 referenced above) - hence this seems best suited for later study. All output RAxML trees (single copy and gene-families) and associated partition schemes are included in our Dryad submission.

Reviewers' Comments:

Reviewer #1:

Remarks to the Author:

Awesome! I'm happy to see this sentence/issue resolved. With that, I have no further revision requests to make.

Congratulations to the authors for their excellent work and publication!

Reviewer #3:

Remarks to the Author:

I am satisfied with the authors answers to my queries and comments.

I only wanted to say that the files they deposited in Dryad are not available, as such is very hard to assess what's in there, formats, info in the files, etc.

EDIT: the files are available now.

REVIEWERS' COMMENTS

Reviewer #1 (Remarks to the Author):

Awesome! I'm happy to see this sentence/issue resolved. With that, I have no further revision requests to make.

Congratulations to the authors for their excellent work and publication!

Thank you for accepting our response.

Reviewer #3 (Remarks to the Author):

I am satisfied with the authors answers to my queries and comments.

I only wanted to say that the files they deposited in Dryad are not available, as such is very hard to assess what's in there, formats, info in the files, etc.

EDIT: the files are available now.

Thank you for accepting our answers to your queries and comments.